# MRL-Based Model for Diverse Bidding Decision-Makings of Power Retail Company in the Wholesale Electricity Market of China

**Ying Wang** [1],*(iD), **Chang Liu** [1], **Weihong Yuan** [1] **and Lili Li** [2]

[1] Key Laboratory of Measurement and Control of Complex Systems of Engineering, Ministry of Education, Southeast University, Nanjing 210096, China
[2] NARI Group Corporation (State Grid Electric Power Research Institute), Nanjing 211106, China
* Correspondence: wyseu@seu.edu.cn

**Abstract:** Power retail companies in the electricity market make profits through buying and selling power energy in the wholesale and retail markets, respectively. Traditionally, they are assumed to bid in the wholesale market with the same objective, i.e., maximize the profit. This paper proposes a multiagent reinforcement learning (MRL)-based model to simulate the diverse bidding decision-making concerning various operation objectives and the profit-sharing modes of power retail companies in China's wholesale electricity market, which contributes to a more realistic modeling and simulation of the retail companies. Specifically, three types of operation objectives and five types of profit-sharing modes are mathematically formulated. After that, a complete electricity market optimization model is established, and a case study with 30 retail companies is carried out. The simulation results show that the proposed method can effectively model the diverse bidding decision-making of the power retail companies, which can further assist their decision-making and further contribute to the analysis and simulations of the electricity market.

**Keywords:** multiagent reinforcement learning; bidding strategy; decision making; electricity market; power retail company; electricity market simulation

## 1. Introduction

Since March 2015, China has issued the plan "Several Opinions on Step Deepening Electricity System Reform No. 9", and a new wave of reform of the power and electricity sectors was launched, which aims at promoting more competition in the generation and retail companies [1]. Considering that China's electricity sector occupies more than 20% of the world's electricity supply, which is the largest one around the world, such a reformulation is relevant globally as many economies are striving to create competitive electricity markets [2]. Power retail companies in the electricity market buy the power energy from power-generating companies in the wholesale market, sell it to the end users in the retail market, and earn the price difference. In the wholesale market, they bid prices/quantities, and the power-generating companies offer prices/quantities; then, the power exchange center will clear the market using an optimization model. In the retail market, they sign contracts with the end users to sell the electricity. As one of the largest human-made physical systems, myriad studies have been conducted on bidding, decision-making, and simulations of the electricity market and its participants. The relevant literature is reviewed in Table 1.

Game theory and optimization-based methods constitute the most widely employed methodological framework for studying market biddings [3]. In ref. [4], the Cournot game model was used to study the bidding behavior of power-generating companies with capacity limitations. In ref. [5], the Stackelberg game model was used to study the bidding strategy of a virtual power plant. The Nash equilibrium is often used as the solution concept,

since participants are noncooperative and each maximizes its own profit [6–9]. In addition, ref. [10] studies the behaviors among the power plants, transmission and distribution companies, and electricity retailers in a cooperative game in China's electricity market. To model the bidding model concerning the market clearing process, bilevel optimization and mathematical programming with equilibrium constraints (MPEC) methods have been widely used and provide rigorous logical tools to study bidding decision-making [11,12]. In these methods, the participants maximize their objectives at the upper-level problem, while the system operation cost is minimized at the lower-level problem. The bilevel problem is usually converted into a single-level mixed-integer linear programming (MILP) problem using the primal-dual formulation with linearized constraints.

Nevertheless, the above methods exhibit several fundamental limitations: Firstly, these methods either focus on the equilibrium derivation or optimization of the bidding strategy for a given participant but fall short of simulating the whole market operation with multiple participants. When studying the diverse bidding decisions of power retail companies and the impacts on the market, the interactions among different participants and their comprehensive effects need to be considered, and thus, the markets with all the participants need to be studied together rather than one-by-one optimization. Secondly, the above approaches often have complicated mathematical formulations, especially for the game-theory-based methods and bi-level optimization problems. For example, they usually solve the problem by merging the subproblem into the main problem under Karush–Kuhn–Tucker (KKT) optimality conditions. However, KKT optimality conditions only work when the subproblem is continuous and convex [13]. However, when considering other participants' diverse bidding models, the models could have several nonconvex conditional functions and some binary variables, making it challenging to conduct such a KKT conversion.

Driven by the rapid advancements in artificial intelligence and machine learning, agent-based reinforcement learning methods are often applied for simulating a large-scale system with distributed decision-making processes [14,15], making them an alternative to game theory and optimization-based methods. Multiagent reinforcement learning (MRL)-based models fully separate the modeling of the participants' bidding (agents) and the market clearing process (environment), and can easily be applied to simulate the whole market with multiple participants. The participants with diverse bidding models can be modeled one by one without considering the others, and the interactions between them will be illustrated during repeated games of bidding. Moreover, the learning process can clearly show the changing trends in the bidding decisions of the participants, making it possible to study the evolution process. In recent years, MRL has been used in the electricity market complex adaptive system (EMCAS) of Argonne National Laboratory (U.S.) [16], GRIDVIEW of ABB [17], agent-based modeling of electricity systems (AMES) of Iowa State University [18], Powerweb of Cornell University [19], and the electric market simulation system of the China Electric Power Research Institute [20], etc.

Regarding the algorithms, classical RL algorithms mainly include Roth–Erev (RE) learning [21], Q-learning [22], deep Q-learning (DQN), deep deterministic policy gradient (DDPG), etc. RE learning has been often used in the simulations of the day-ahead electricity market [23], forward electricity market [24], long-term electricity market [25], etc. Reference [26] applied RE to simulate the bidding decision-making of generators in the demand response market with commercial buildings. In [23], generation companies are simulated by a variant of RE learning concerning the power flow constraints. In [27], RE learning and its variants are applied to simulate the bidding behaviors of distributed generations in the electricity market. Reference [28] applies Q-learning and its variants to simulate the electricity markets. In [29], the particle swarm optimization algorithm and Q-learning are combined to simulate the electricity market. In [30], a bilevel mathematical optimization problem and Q-learning model are combined to maximize the profit of the individual market players. Recently, with the development of deep learning methods, references [31–33] applied DDPG and DQN to electricity market simulation. In [33], DQN

is applied to study generational bidding while maximizing average long-term payoffs. Ref. [34] proposes a combined deep Q-learning and transfer-learning method to simulate the electricity market. In [35], deep RL is used to study the equilibrium point of the electricity and the overall energy efficiency of the power network. In [36], deep Q networks and DDPGs were combined to study approximating Nash equilibrium in the day-ahead electricity market.

**Table 1.** Literature review.

| Reference | Bidder | MRL-Based | Learning Algorithm | Bidding Objective |
|---|---|---|---|---|
| [3–12] | GenCo and/or RetCo | No | None | Max. profits |
| [15] | GenCo | Yes | Not mentioned | Max. profits |
| [16–20,23–27] | GenCo/RetCo /microgrids/ distributed energy/ demand | Yes | RE and its variants | Max. profits or min cost |
| [28–30] | GenCo | Yes | Q-learning and its variants | Max. profits |
| [31–36] | GenCo | Yes | Deep RL | Max. profits |
| proposed | RetCo | Yes | RE | Diverse |

Abbr: GenCO: generation company; RetCo: retail company; EV electric vehicle.

In China, power retail companies have features of larger numbers, smaller capacities, and more flexible operations [10]. Based on the annual report from the Guangdong power exchange center company [37], the number of retail companies is 507, which is much more than that of generation companies (188). Meanwhile, the HHI (Herfindahl–Hirschman Index) is 1295 for the generation company and 376 for the retail company, showing that the market concentration of retail companies is much lower than on the generation side. As to the bidding strategies of retail companies, their bidding decisions are diverse and affected by many factors [38,39]. Reference [10] summarizes the retail companies into four types based on their ownerships; different types of retail companies have different cost and income sources, and finally form different and diverging cooperative game structures. Reference [38] analyzes the bidding results of the retail companies in the Guangdong province and concludes that the retail companies with different ownerships have large differences in their operations. Reference [39] finds that the main factor that affects the profitability of retail companies is the amount of electricity, and the spread of rebate modes between the retail companies and the end users also has significant impacts on their profit. In addition, some retail companies, especially the new participants or the ones that have large enterprises as shareholders, often have diverse operation objectives in the starting period, such as expanding their market share or improving their reputation. One of the reasons accounting for such losses is that they often bid at a high buying price to guarantee their bids can be cleared without concern for profit issues and would rather have low profits or lose money to keep a solid connection with the customers and expand their market share. Observing the profit data in [37], 42% of retail companies have negative annual profits. China's National Energy Administration (NEA) finds that retail companies burn money to maximize their trading electricity amount, resulting in prices falling into an irrational range, and appeals to the retail companies to be rational in the fierce competition [40]. China Weekly has investigated the operations of the electricity retail companies, finding that they can make a profit only after acquiring enough end users, and their operations are somewhat similar to 'circling fans' of the internet companies, i.e., getting a certain amount of 'fans' at first and then obtaining the flow of cash [41]. As a result, they often bid irrational prices to ensure they can win, and these irrational and diverse bidding decision-makings will lead to the market operation being highly different and intractable.

From the above review, it can be seen that there remains a gap regarding the modeling method of the irrational and diverse bidding decision-making of retail companies. Most of the existing work focuses on modeling the generation companies' biddings or improving the learning process with fancy algorithms, while the objective models of retail companies are mostly assumed to be maximizing profits or minimizing costs with perfect rationality. Such an assumption often holds for generation companies [3–36]; however, it is inconsistent

with our observations of the retail companies in the real electricity market. To the best of our knowledge, there are still no specific studies focusing on modeling the diverse bidding decision-making of retail companies. In addition, the profit-sharing modes of the retail companies with the end users have a large impact on the retail company's bidding decisions in the wholesale market, but these modes have not been studied in the previous work. To fill this gap, this paper first proposes an MRL-based model for diverse bidding decision-making of power retail companies in the wholesale electricity market, which extends the bidding objective from a traditional single one to several diverse ones. Meanwhile, the profit-sharing modes are mathematically modeled. The main contributions of this paper are as follows:

1.  Propose a model for diverse bidding decision-making among the power retail companies in China's electricity market. Different from the existing work only considering the profit-maximizing objective, we first build up the mathematical models of diverse bidding decision-making concerning three types of operational objectives and five types of profit-sharing modes, which contribute to a more realistic simulation and modeling of retail companies.
2.  An MRL-based electricity market simulation model with power retail companies' diverse bidding decision-making is established to capture individual subjectivity and irrationality in the bidding decision-making, which contributes to a more realistic simulation and modeling of the electricity market.

The proposed model can help market participants predict the different bidding results under different bidding decisions, as well as predict the opponents' bidding decisions. It can also equip the market operators with a more realistic simulation tool, which may further help to discover a more realistic evolution trend, avoid some operation risks, and maintain a more stable market status. From the perspective of the policymakers, such a simulation platform can help to unfold the possible consequences brought by a certain policy, which could be used as a tool to verify the effectiveness of the market rule.

The remainder of this paper is organized as follows: Section II introduces the market structure and clearing mechanisms of the proposed method. Section III presents the diverse bidding decision-making models of the power retail companies. Section IV presents the MRL-based electricity market simulation model with diverse bidding decisions of the power retail companies. In Section V, we carry out a case study to verify the effectiveness of the proposed method, and Section VI presents the conclusions.

## 2. Bidding in Wholesale Electricity Market

### 2.1. Market Structure

The power industry in China was restructured and reformed in March 2015, aiming to build a comprehensive and efficient power market system. Currently, nearly all provinces have implemented policies to promote the electricity market. From the viewpoint of the market types, the typical constructed markets include a forward market (yearly, monthly, and intramonthly), an ancillary service market (frequency regulation, energy reserve, peak regulation, etc.), a spot market (day-ahead and real-time), and a capacity market [1]. From the viewpoint of the trading methods, the markets can be divided into bilateral negotiations, market auctions, and listed transactions [42]. From the viewpoint of the participants, the typical markets include the wholesale markets and the retail markets. In the wholesale market, generation companies compete to sell electricity directly to retail companies and large consumers, and the trading is conducted in the power exchange center, while the national grid company is responsible for the power dispatch, transmission, and distribution. In the retail market, retail companies can participate in the market on behalf of consumers.

Generally, retail companies buy electricity from the power generation companies in the wholesale market and sell electricity to the end users in the retail market to earn the price difference, as shown in Figure 1. The wholesale market refers to the electricity market between power generation companies and retail companies. The electricity retail market refers to the market between retail companies and end users.

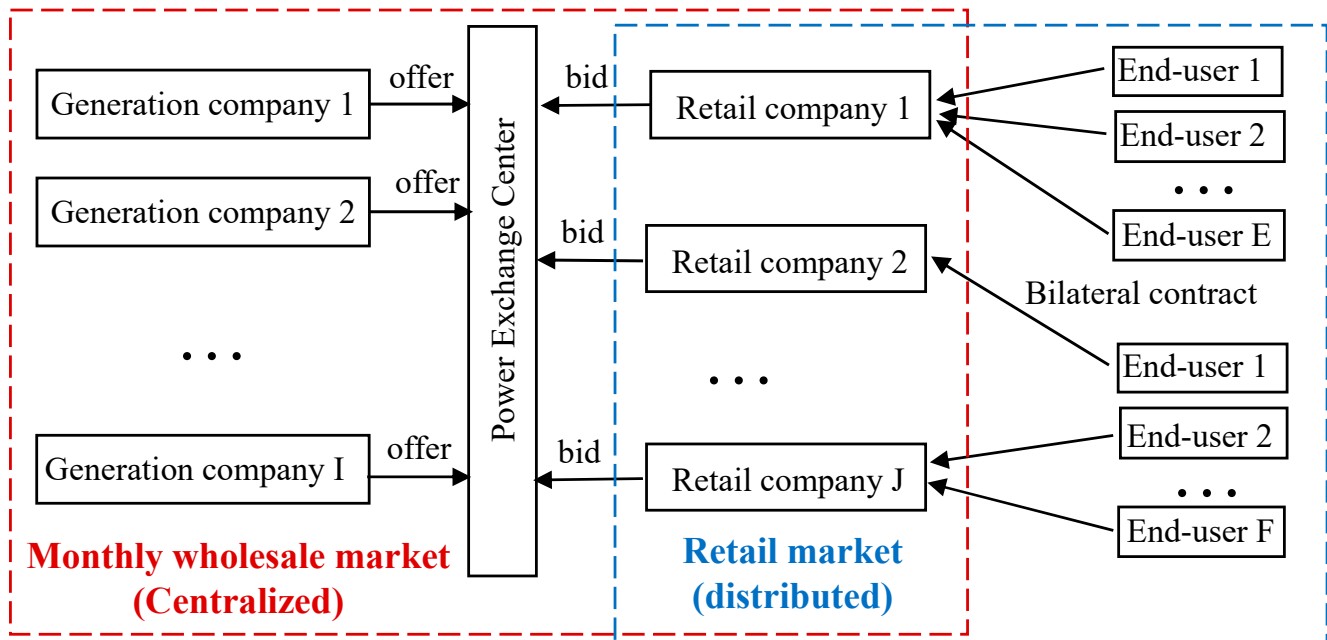

**Figure 1.** Structure of the electricity market.

The electricity wholesale market can be divided into yearly market, monthly market, day-ahead market, and real-time market from the view of the timeline [43]. In the yearly wholesale market, trading is often processed distributionally by bilateral negotiations, which is called the bilateral market. In the bilateral market, when the generation company and the retail company reach an agreement on the trading, they will sign a contract. In monthly, day-ahead, and real-time wholesale markets, trading is often processed centrally by market operators in the power exchange center, and such a market is called the centralized market.

In this paper, we focus on the power retail companies' bidding decision-making in the monthly wholesale market. Considering that the markets and regulations for power retail companies and the countries in which they operate are not the same in each country of the world and are even different in different provinces in China, three assumptions need to be made for our models.

1. The impacts of the yearly market and spot market on the monthly market bidding decisions are not considered to simplify the problem formulation, which is commonly applied in most relevant studies.
2. The impacts of the trading between the power retail companies and end users are considered when building up the profit-sharing modes. In fact, the direct and indirect impacts from the end user's side could be complicated, and trading forms differ significantly worldwide, which will all have impacts on the decision-making in the wholesale market. In the U.S., the end users often pay bills to the retail companies through a power purchase agreement (PPA) or electricity retail plan (ERP) [44], but the prices are usually independent of the wholesale market. Different from the U.S., many retail companies and the end users in China have an agreement on how to divide the profit from the wholesale market, which is called the spread rebate mode in [39]. To extend it to a more general model, this paper builds up a mathematical model of the profit-sharing modes.
3. The impacts of the energy deviation settlement mechanism (EDS), deviation mutual insurance (DMI), contract transfer market, and intramonth market are not included in this paper. In China's monthly market, the above mechanisms have been applied in some markets to settle and reduce the deviation between consumption and energy contracts of the retail companies [45] and may have further impacts on the bidding

decisions on the wholesale markets. In this paper, these impacts are not considered to simplify the problem formulation.

## 2.2. Market-Clearing Mechanism

In the wholesale market, retail companies usually have options to bid on a decremental price-quantity curve, which is divided into several price-quantity segments [46]. Figure 2a,b illustrate the bidding curves with 1-segment and 3-segment bidding rules, respectively. Under the multisegment bidding rule, a retail company can divide its available required quantity into several segments and bid on those segments at monotonically decreasing prices. The number of bidding segments varies in different markets. For example, in the Australian electricity market [47] and PJM (Pennsylvania-New Jersey-Maryland Interconnection) market in the U.S. [48], retail companies can bid at most 10-segment bidding curves. In Jiangsu Province of China, the bidding curve must be one segment, while 3-segment and 6-segment biddings are allowed in Guangdong and Zhejiang provinces, respectively. In this paper, the 3-segment bidding rule is applied, while the models also apply to other 1-segment or multisegment markets.

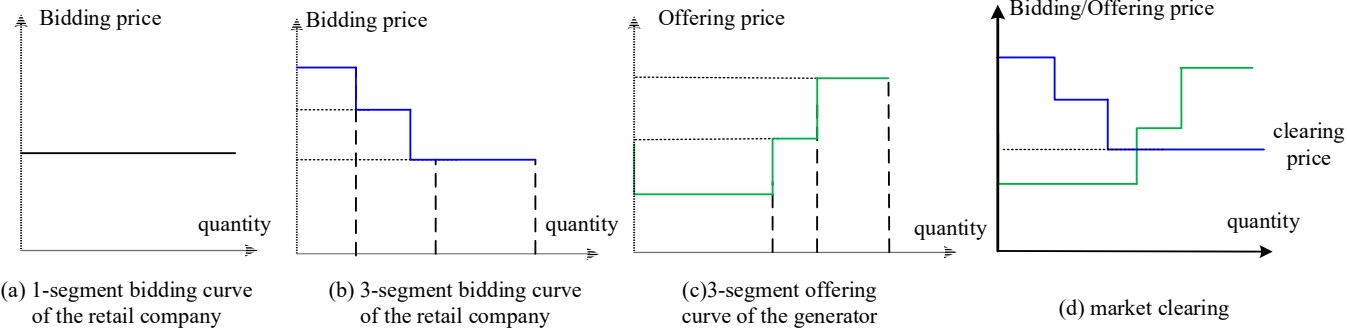

(a) 1-segment bidding curve of the retail company

(b) 3-segment bidding curve of the retail company

(c)3-segment offering curve of the generator

(d) market clearing

**Figure 2.** Illustrative figure of the market clearing.

The clearing mechanism of the market is the core of the electricity market. At present, there are two main methods for clearing in the electricity market: the pay-as-bid (PAB) or pay-as-clear (PAC) pricing scheme in the market, and the PAC is also called a uniform price scheme [49]. In the PAB market, the winning participants are paid at their respective bidding prices, whereas all winning participants are paid at the same market clearing price in the PAC market [50]. This paper adopts the PAC pricing scheme, which is also the most common clearing mechanism adopted for all wholesale electricity markets in the U.S. [51] and most markets in China [52]. When neglecting the power flow constraints, the market arranges the prices of retail companies from high to low, and the prices of generators are arranged from low to high. Finally, the price is cleared according to the intersection price, as shown in Figure 2c,d. Note that all the bidding segments will be cleared at the same market price according to the PAC mechanism.

## 3. Modeling of Diverse Bidding Decision-makings

### 3.1. Three-Dimension Influencing Factors

Before modeling the diverse bidding decision-making process, the influencing factors of the bidding strategies of the retail companies are identified as the followings: market share, operation objectives, and profit-sharing modes, as shown in Figure 3.

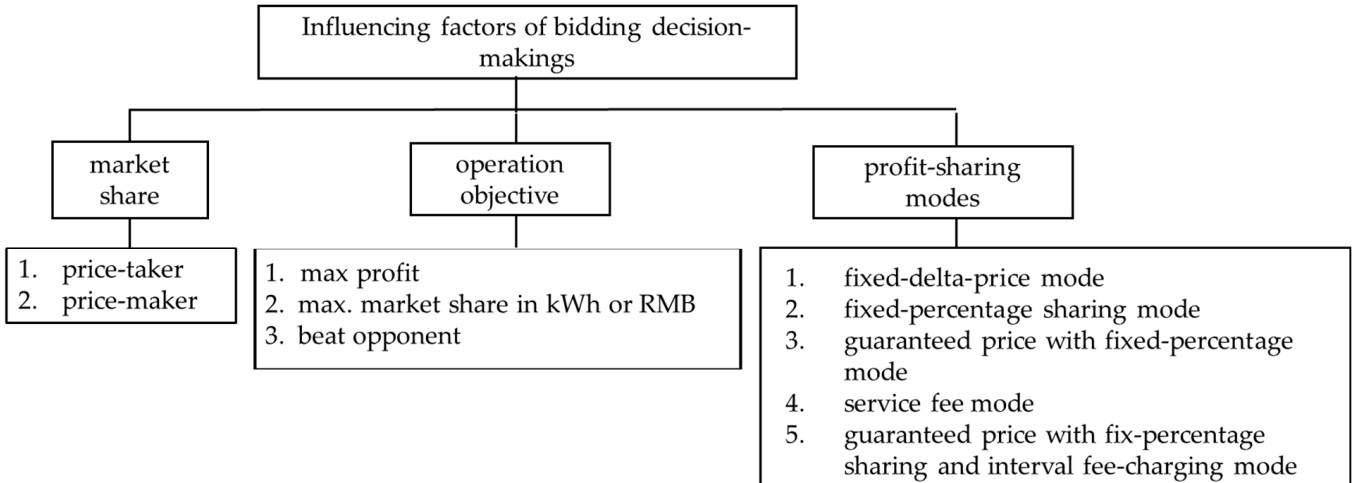

**Figure 3.** Influencing factors of bidding decision-makings.

1.  Market share. From the perspective of whether the influences of biddings/offers on clearing prices are considered, a participant in the market can be generally defined as a price-taker or price-maker depending on its market share [53]. If a participant is small-scale and its bidding/offering actions can hardly impact the market clearing price, it is called a price taker. Under the pay-as-clear market mechanism, a price-taker generator often bids the lowest price, i.e., the "floor price", and the price-taker retail company often bids the highest price, i.e., the "ceil price", which ensures that the bids/offers can be cleared successfully. On the contrary, price makers will often take more complex bidding strategies in the market bidding. Therefore, we assume that price takers bid at stable prices and that price makers use the MRL algorithm to optimize their bidding decisions.

2.  Operation objective. The operation objective determines the objective function model in retail company modeling. From a rational point of view, the market participants should maximize the profits, but observations from the real markets seem to not obey this rule. Especially in a newly developing market, e.g., China, power retail companies have diverse operation objectives and often adopt unusual bidding strategies. For example, some retail companies are established by generation power companies. They are well capitalized, would rather lose money to expand sales of electricity, and often fight the price war. Another example is that some retail companies have large enterprises as shareholders, such as Alibaba, and often bid at a high price to guarantee their bids can be cleared, while selling the electricity at a low price to maintain a stable connection with the end users. Through investigation and analysis of the actual bidding results from news and reports, we divided their operation objectives into three categories, as shown in Figure 3. The specific models will be expanded in Section 3.2.

3.  Profit-sharing modes in the bilateral contracts with the end users. In China, if an end user does not participate in the market by themselves or through a retail company, they will pay the electricity fee at a default fixed price, which is called regulated price [54]. Regulated prices are set by the provincial National Development and Reform Commission in China, as shown in Table A1. When an end user buys electricity through a retail company, the cost-savings, in total, are the price difference between the regulated price and the market clearing price. Such a profit induced by the price-in-difference will not be totally obtained by either the retail company or the end users, in fact. Based on our observation, they will redistribute the total profits according to different methods, which have been previously negotiated and written into their bilateral contracts. In this paper, we name this redistribution "profit-sharing mode". For example, they will set a fixed percentage to redistribute the total profit, which

is called *fixed-percentage mode*. To avoid such a loss, some end users will require the power retail company to guarantee a trading price that is less than the regulated price, which is called a *guaranteed price in fixed-percentage mode*. Note that the profit-sharing modes could be very complicated in reality. For instance, when the clearing price is higher than the regulated price, the total profit is negative. The detailed model will be presented in Section 3.3.

### 3.2. Diverse Operation Objectives Modeling

In a realistic electricity market, it is often observed that not all the participants bid rationally to maximize profit. In this section, we divide the operational objectives of retail companies mainly into three types.

1.  Maximize profit. The profit can be expressed by the difference between income and cost, as shown in (1). As to the retail company, the cost is for purchasing electricity from the wholesale market at the market clearing price, and the cost of buying the electricity is shown in (2). Generally, the income is from the end users, which can be calculated as the production of the selling price and the selling quantity, as shown in (3).

$$F_s^{\text{obj}} = F_s^{\text{profit}} = F_{s'}^{\text{income}} - F_s^{\text{cost}} \tag{1}$$

$$F_s^{\text{cost}} = q_s^{\text{clc}} \cdot \lambda^{\text{clc}} \tag{2}$$

$$F_s^{\text{income}} = \sum_k q_{s,k}^{\text{contract}} \cdot \lambda_{s,k}^{\text{contract}} \tag{3}$$

where $F_s^{\text{profit}}$ means the operating objective of retail company s is to maximize profit; $F_{s'}^{\text{income}}$ and $F_s^{\text{cost}}$ are its income and cost, respectively; $q_s^{\text{contract}}$ and $\lambda_s^{\text{contract}}$ are the selling quantity and price signed in the contract by the retail company s with the user, respectively; $q_s^{\text{clc}}$ is the cleared quantity of retail company s; $\lambda^{\text{clc}}$ is market clearing price; $k$ means multiple end users.

It should be noted that the profit and cost functions of different retail companies are different. For example, some retail companies are established by large-scale end users, which helps them bid in the market and reduce costs. In this situation, the profit function is shown in Equation (4). Some retail companies construct distributed power grids and make a profit by charging end users a grid fee for grid usage. Therefore, the larger the amount of the electricity transmitted through its distribution power grid, the higher the profit, and the operation objective function can be written as Equation (5). In addition, some retail companies will provide the end users with extra services, e.g., distribution power grid maintenance, to make more profit.

$$F_s^{\text{profit}} = q_s^{\text{clc}} \cdot \lambda_s^{\text{reg}} - F_s^{\text{cost}} \tag{4}$$

$$F_s^{\text{profit}} = q_s^{\text{clc}} \cdot \lambda_s^{\text{gridfee}} \tag{5}$$

where $q_s^{\text{clc}}$ is clearing quantity; $\lambda_s^{\text{reg}}$ is regulated price; $\lambda^{\text{clc}}$ is market clearing price and $\lambda_s^{\text{gridfee}}$ is grid-fee.

2.  Market expansion in kWh or RMB. Some power retail companies will tend to make bidding decisions to guarantee that the electricity can be bought successfully, avoiding violating the contracts signed with the end users, and further avoiding disconnecting with the customers. They often give priority to winning the bid and expanding the market share over making a profit in the short term. The operation objectives of maximizing market share in kWh are provided by Equation (6). When maximizing the market share in RMB, the objective is modeled as (7).

$$F_s^{\text{obj}} = q_s^{\text{clc}} \tag{6}$$

$$F_s^{\text{obj}} = q_s^{\text{clc}} \cdot \lambda_s^{\text{clc}} \tag{7}$$

Note that the above expansion strategies often require maintaining an acceptable level of losses or meeting a basic profitability target to maintain the company's basic operations. When needed to maintain an acceptable level of losses, the operating objective can be described by Equations (8) and (9) with a penalty term.

$$F_s^{\text{obj}} = q_s^{\text{clc}} \cdot \lambda_s^{\text{clc}} - \pi_s^{\text{penalty}} \tag{8}$$

$$\pi_s^{\text{penalty}} = \begin{cases} 0 & if \ F_s^{\text{profit}} \geq F_s^{\text{goal}} \\ \delta_{penalty} \cdot \left( F_s^{\text{goal}} - F_s^{\text{profit}} \right) & if \ F_s^{\text{profit}} < F_s^{\text{goal}} \end{cases} \tag{9}$$

where $\delta_{penalty}$ is the conversion coefficient of the penalty term; $F_s^{\text{goal}}$ is the basic profitability target.

3. Defeat the opponent. From the observation of China's real electricity market, there are some retail companies whose main objective is to defeat a specific opponent. Such companies will know the bidding price of their rivals through their mysterious information source and then bid a slightly higher price than the rival company to ensure that they can win the bid with a greater probability than the rival company. Such a bidding model is given as Equations (10) and (11).

$$F_s^{\text{obj}} = q_s^{\text{clc}} \tag{10}$$

$$\lambda_s^{\text{bid}} \geq \lambda_{aim}^{\text{bid}} \tag{11}$$

where $\lambda_{aim}^{\text{bid}}$ is the bidding price of an opponent company.

### 3.3. Diverse Profit-Sharing Modes

In the electricity market, the profit-sharing modes between the power retail companies and end users have significant influences on the bidding decisions of power retail companies in the wholesale market. The retail company and the end users will redistribute the total profits, which have been agreed upon in advance in the contract. In this paper, five profit-sharing models are identified based on our observations in the real electricity market, and the mathematical model of each mode is proposed. The illustrations of the profit-sharing modes are shown in Figure 4.

1. Fixed-delta-price mode. Power retail companies sign fixed-delta-price contracts with users. The signed price is usually the regulated price minus a fixed price, which is called the delta price. In the industrial field, it is called the fixed-delta-price mode, as given in (12). Under this mode, the end users do not need to take the risk of market price fluctuations and can buy electricity at a fixed price. If the market clearing price is lower than the market clearing price, the retail company can make a profit, and the user does not need to bear the risk of market fluctuations. A higher delta price indicates a higher profit for the end user and a lower profit for the retail company.

$$F_s^{\text{profit}} = q_s^{\text{contract}} \cdot \left( \lambda_s^{\text{reg}} - \lambda^{\text{clc}} - \Delta\lambda_s \right) \tag{12}$$

where $\Delta\lambda_s$ is the delta price.

Note that they use fixed-delta-price rather than fixed-price because the regulated prices for each retail company are different even if they are in the same wholesale market.

2. Fixed-percentage sharing mode. The retail company and the end users agree on a sharing percentage of the total profit. After the market is cleared, the total profit is proportionally divided according to this value, as given in (13). Under this model, the two parties are more affected by market price fluctuations.

$$F_s^{\text{profit}} = q_s^{\text{contract}} \cdot \left( \lambda_s^{\text{reg}} - \lambda^{\text{clc}} \right) \cdot \theta \tag{13}$$

where $\theta$ is the fixed percentage.

3. Guaranteed price with fixed-percentage mode. This mode is a combination of fixed-delta-price and fixed-percentage sharing modes. The retail company and the end users will agree on a guaranteed price and a fixed percentage. When the clearing price is higher than the guaranteed price, the end users will pay the electricity bill to the retail company at the guaranteed price, and the retail company will suffer the loss. When the clearing price is lower than the guaranteed price, the retail company and the end users will share the profit part, which is between the clearing price, and the guaranteed price, at the agreed fixed percentage. Assuming that the guaranteed price is $\lambda_s^{\text{base}}$, the percentage of profit-sharing is $\theta$, the profit-sharing mode is as given in (14).

$$F_s^{\text{profit}} = \begin{cases} q_s^{\text{contract}} \cdot \left( \lambda_s^{\text{base}} - \lambda^{\text{clc}} \right) & if \lambda^{\text{clc}} \geq \lambda_s^{\text{base}} \\ q_s^{\text{contract}} \cdot \left( \lambda_s^{\text{base}} - \lambda^{\text{clc}} \right) \cdot \theta & if \lambda^{\text{clc}} < \lambda_s^{\text{base}} \end{cases} \tag{14}$$

It should be noted that the guaranteed price with a fixed-percentage sharing mode is more beneficial to the end users, while the power retail companies have the risk of loss when the clearing price is higher than the guaranteed price. In contrast, there is another type, which is the reverse mode of "guaranteed price with fixed-percentage sharing mode", which guarantees the profit of the retail company rather than the end users.

4. Service fee mode. Under this mode, when winning the bidding, the retail company will charge a service fee according to the clearing quantities. The profit model is given as (15).

$$F_s^{\text{profit}} = q_s^{\text{contract}} \cdot \lambda_s^{\text{service}} \tag{15}$$

where $\lambda_s^{\text{service}}$ is the service fee per kWh.

5. Guaranteed price with fixed-percentage sharing and interval fee-charging mode. The retail company and the end users agree on the guaranteed price and the fee-charging price in the contract, while the interval between the guaranteed price and the fee-charging price is called the fee-charging interval. When the clearing price is higher than the guaranteed price, the retail company will suffer a loss. When the clearing price is higher than the fee-charging price but lower than the guaranteed price, the retail company will charge the service fee to the end users. When the clearing price is lower than the fee-charging price, the extra profit is shared at the fixed percentage. The bidding model is given as (16).

$$F_s^{\text{profit}} = \begin{cases} q_s^{\text{contract}} \cdot \left( \lambda_s^{\text{base}} - \lambda^{\text{clc}} \right) & if \lambda^{\text{clc}} \geq \lambda_s^{\text{base}} \\ q_s^{\text{contract}} \cdot \lambda_s^{\text{service}} & if \lambda_s^{\text{fee}} \leq \lambda^{\text{clc}} < \lambda_s^{\text{base}} \\ q_s^{\text{contract}} \cdot \left[ \lambda_s^{\text{service}} + \left( \lambda_s^{\text{fee}} - \lambda^{\text{clc}} \right) \cdot \theta \right] & if \lambda^{\text{clc}} \leq \lambda_s^{\text{fee}} \end{cases} \tag{16}$$

where $\lambda_s^{\text{fee}}$ is the price of the charging fee.

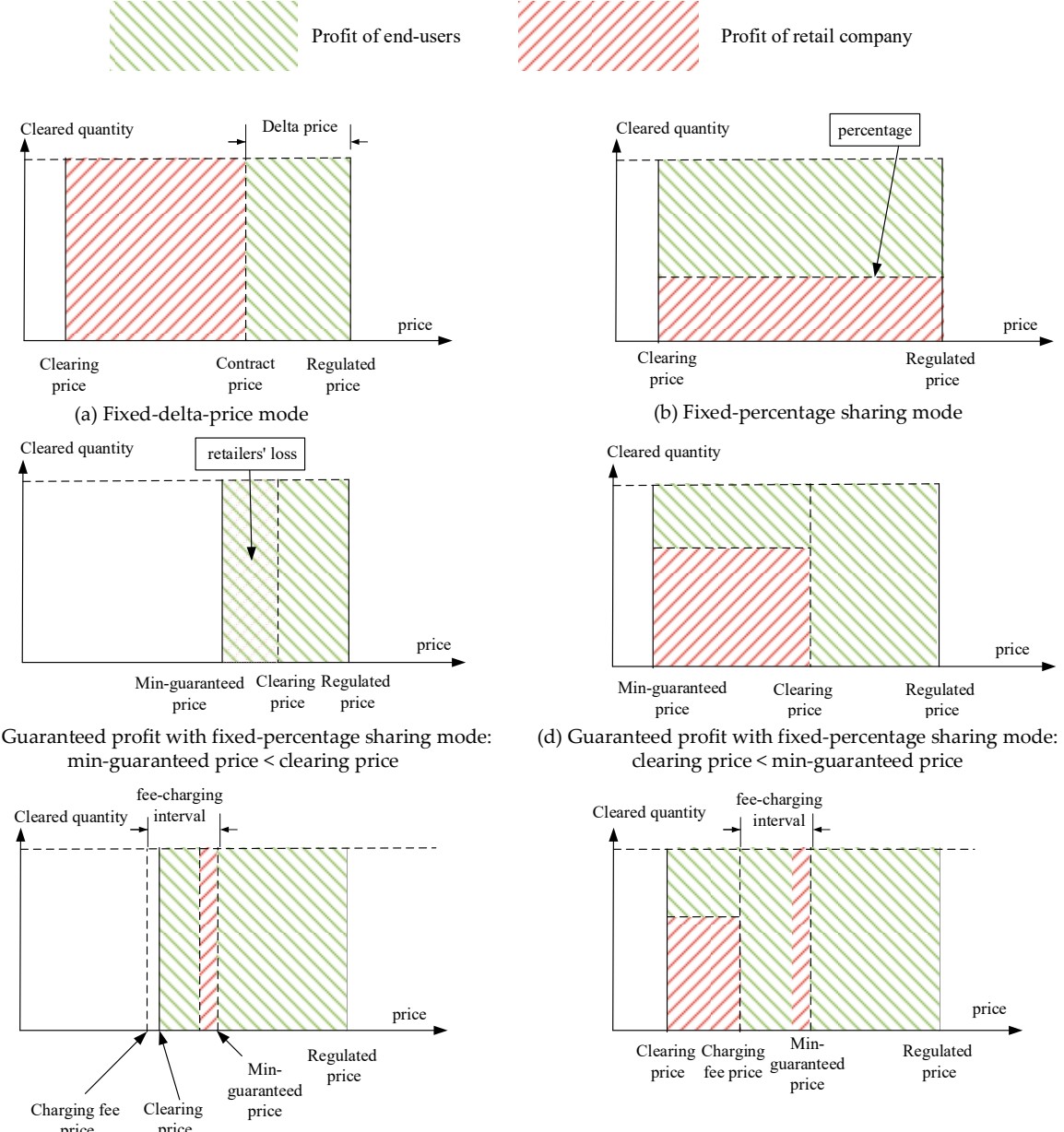

**Figure 4.** Illustrative figures of profit-sharing modes.

## 4. MRL-Based Electricity Market Simulation Model

### 4.1. MRL-Based Electricity Market Simulation

Reinforcement learning belongs to the field of machine learning, and the most important feature is the mutual restraint between the learning agent and the environment. The agent optimizes the decision-making behavior through the reward and penalty signals of the environment. In the electricity market, MRL is mainly used to simulate the decision-making behavior of market members when participating in market bidding. Market members optimize their next decision-making strategy through the return of each bidding information. The Roth–Erev algorithm [21] is a classical algorithm in the field of electricity market simulation that was proposed by Roth and Erev in 1998, was originally used by the AMES, and has since been widely used in the simulations of many electricity markets. In this work, we select the Roth–Erev learning algorithm to simulate the electricity

market, which decides and optimizes the next action based on historical data and recent learning experiences. Compared to other algorithms, the Roth–Erev algorithm is simple to apply without designing a delicate Q-table or building up complicated deep networks, and the algorithm itself is not the focus of this paper. Therefore, this paper applies RE to simulate the proposed model.

MRL consists of multiple agents and an external environment. Agents get feedback through their own decision-making information, and in the decision-making environment, the learning process is shown in Figure 5. After the agent takes an action, if the external environment feedback is that it is a positive benefit, the decision's selection propensity value is strengthened, and the selection probability will increase the next time the decision is made. After the agent has gone through several rounds of learning, the propensity value of a certain choice or several choices keeps accumulating and eventually converges.

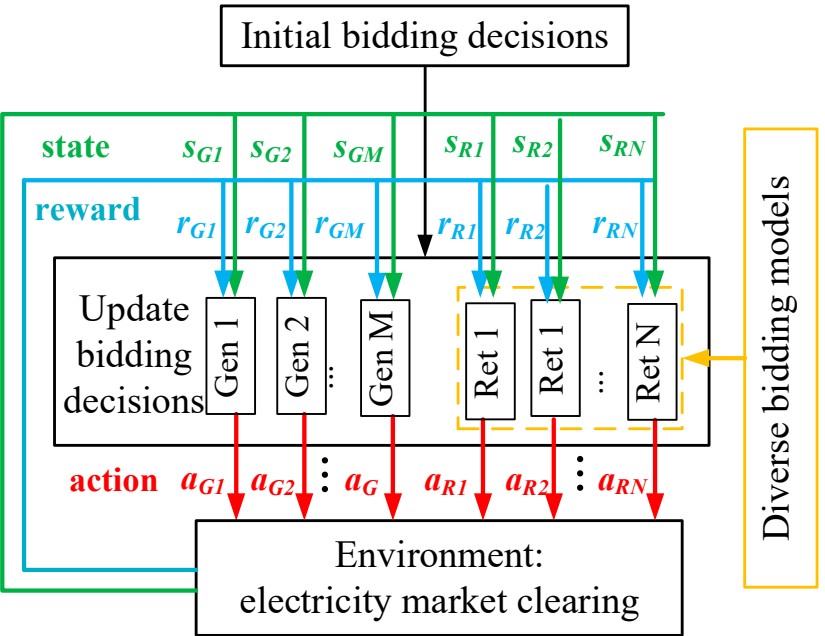

**Figure 5.** Electricity market simulation model based on reinforcement learning.

Considering that the generation companies' biddings are not the focus of this work, we set stable bidding strategies for the generation companies, while $\lambda_{g,j}^{\text{offer}}$ and $q_{g,j}^{\text{offer}}$ are the offering price and quantity of the generation company $j$, respectively.

This paper adopts the PAC pricing scheme, which is also the most common clearing mechanism. When neglecting the power flow constraints, the market arranges the prices of retail companies from high to low, and the prices of generators are arranged from low to high. Finally, the market clearing optimization model is provided by Equations (17)–(20). After the market is cleared, the clearing price can be derived from the dual variable of Equation (18).

$$\max \sum_i \lambda_{s,i}^{bid} q_{s,i}^{bid} - \sum_j \lambda_{g,j}^{\text{offer}} q_{g,j}^{\text{offer}} \tag{17}$$

$$\sum_{i,n} q_{s,i}^{clc} = \sum_j q_{g,j}^{clc} \tag{18}$$

$$0 \le q_{s,i}^{clc} \le q_{s,j}^{bid} \tag{19}$$

$$0 \le q_{g,j}^{clc} \le q_{g,j}^{\text{offer}} \tag{20}$$

where $\lambda_{s,i}^{bid}$ and $q_{s,i}^{bid}$ are the bidding prices and quantities of the retail companies, $i$.

### 4.2. Reinforcement Learning-Based Modeling of the Retail Company

In the simulation of the electricity market, market participants can be regarded as intelligent agents. Each agent chooses its own bidding strategy for decision-making, and the market returns the income to the agent after certain rules are calculated. After calculating according to certain rules of the market, the income is fed back to the agent. Each agent updates the bidding tendency through the reward. Agents need to continue to explore the external environment by learning. In the simulation of the electricity market, the external environment is constantly changing, so the agents need to calculate the reward after long-term execution to evaluate the action strategy. Each agent optimizes decision-making behavior by using historically accumulated experience tendencies and recent learning experiences to learn the bidding strategies with the largest average reward.

The process of MRL learning includes the following processes: constructing market members' bidding strategies, selecting strategies according to trends, calculating rewards, updating the propensity values of each strategy, and looping to get the best strategy. The following are the algorithm steps, and the flow chart is shown in Figure 6.

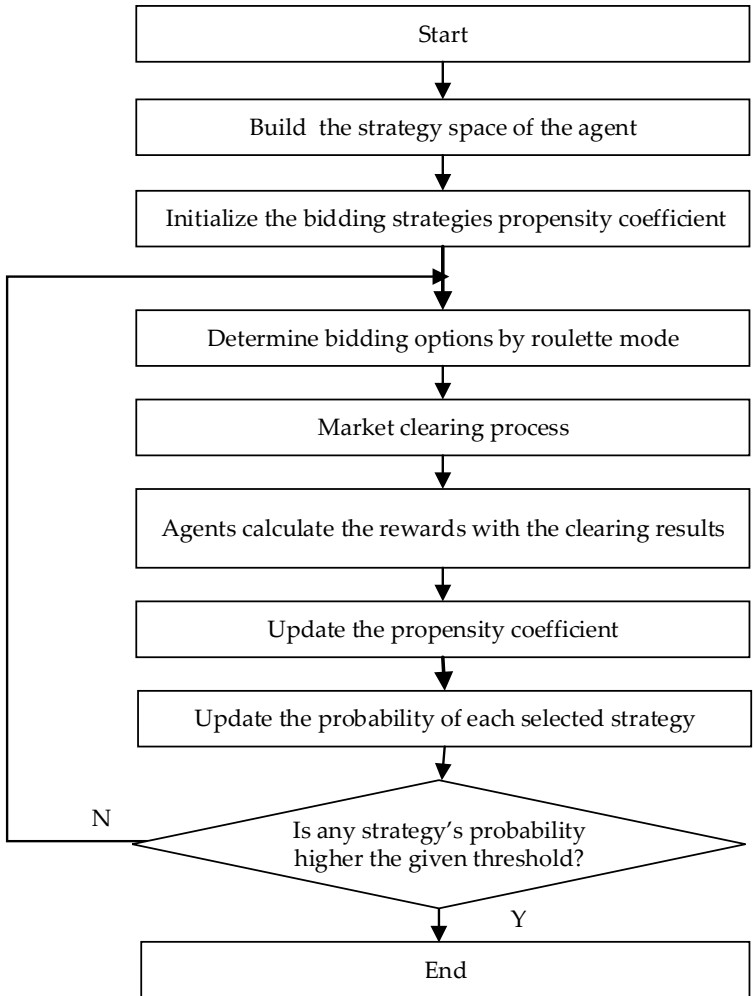

**Figure 6.** Simulation flow chart.

1. Constructing market participants' bidding strategies. For the electricity retail companies, the bidding strategy set is $A_s = \left\{ a_{s,1}, a_{s,2} \cdots a_{s, SN_{s,i}} \right\}$, where $SN_{s,i}$ is the number of bidding strategies of the retail companies, $i$.

2. Select strategies according to trends. When the decision maker chooses a strategy, it is selected according to the propensity value of each strategy. The greater the propensity value of the strategy, the higher the probability of being selected. Then, according to

roulette, select the strategies. Assume that the retail company $i$ has $SN_{s,i}$ bidding strategies, and the initial probability of each strategy is $1/SN_{s,i}$. The decision-maker randomly selects a strategy as the initial strategy based on each probability.

3. Calculating profit as the reward value. The retail company submits the bidding prices and quantities to the power exchange center. The power exchange center clears the market with the optimization model of Equations (17)–(20), and calculates the clearing price. Each participant calculates the reward value of this bidding according to their own operation objectives and profit-sharing modes.

4. Updating the propensity values of each strategy. In the Roth–Erev algorithm, each strategy corresponds to the decision maker's behavioral propensity value $Q$ and selection probability $P$ as the basis for agent decision-making. Each strategy updates the behavior propensity value $Q$ according to the feedback. Suppose that in round $t$, based on the function value $F_{s,i,p}^{\text{obj},t}$, which is obtained from the clearance, the retail company chooses the $p$-th strategy $a_{i,p}$. Then, in the round t + 1, update the behavior propensity value corresponding to each strategy, as given in Equation (21):

$$
Q_{s,i,p}^{t+1} = \begin{cases} (1-r)Q_{s,i,p}^{t} + (1-e)F_{s,i,q}^{\text{obj},t} & if \ \ p = q \\ (1-r)Q_{s,i,p}^{t} + \dfrac{e \cdot Q_{s,i,p}^{t}}{SN_{s,i}} & if \ \ p \neq q \end{cases} \tag{21}
$$

where $i$ is the retail company $i$, $t$ is the bidding round; $m$ is the strategy number; $SN_{s,i}$ is the number of strategies of the retail company $i$; $r$ is the forgetting factor; $e$ is an empirical parameter, and $q$ is the selected strategy number.

According to the behavior propensity values, update the selection probability of each strategy, as given in Equation (22). $c_s^t$ is the cooling parameter, which is dynamically adjusted according to the propensity values. $\varepsilon$ is a real number that is greater than 0, as given in Equation (23).

$$
prob_{s,i,p}^{t+1} = \frac{\exp\left(Q_{s,i,p}^{t}/c_s^t\right)}{\sum_{p=1}^{SN_{s,i}} \exp\left(Q_{s,i,p}^{t}/c_s^t\right)} \tag{22}
$$

$$
c_s^t = \frac{\varepsilon}{SN_{s,i}}\sum_m Q_{s,j,p}^t \tag{23}
$$

Loop to get the best strategy. Repeat the calculation many times until you get a strategy selection probability >0.99 or reach the round number limit.

## 5. Case Study

### 5.1. Case Settings and Data

The section conducts a simulation case study of an electricity market concerning the diverse bidding decision-making of power retail companies. Since the detailed bidding data of the retail companies is usually considered confidential information in most practical markets, and thus, it is not made public due to data privacy issues and data security concerns. In this work, we design a virtual case in which the main parameters are set based on the limited public information of the monthly market of Guangdong Province in China [55]. The participant numbers, the total amount of biddings, and the market clearing prices will be open to the public one year after the trading month.

In this work, the designed market includes 10 generation companies and 30 retail companies, where the ratio of the numbers is overall consistent with the actual numbers in the real market, but the values are reduced for the sake of simplifying the simulation work. The detailed parameters are shown in Table A2. In order to compare the results with and without diverse bidding models, the environments need to be kept the same. Meanwhile, considering that there are three types of objectives, five types of profit-sharing modes, and different parameters in both models, the number of combinations of the different factors is large. In this work, the case is well-designed to compare the impacts of a one-by-one

parameter of the retail companies while keeping the market environment exactly the same. Specifically, 30 retailers were well-designed in pairs, and most parameters and settings of the retail companies in the same pair were the same so that the impacts arising from the different parameters could be exposed.

The total bidding electricity of participants was 6 billion kWh (generation company) and 5.45 billion kWh (retail company), respectively, and the supply-demand ratio was 1.1. Meanwhile, to be consistent with the multisegment bidding rules in the realistic market, this case applies a 3-segment bidding rule. The bidding quantities of the retail companies with MRL training are set as 1/3 of the total quantity. The bidding prices of the first and second segments are predetermined as the ceiling prices of the market (0.45 RMB/kWh), or their contact prices with the end users. The MRL trains the bidding prices of the third segment, but all three segments will be cleared at the same market price according to the market clearing rule.

### 5.2. Bidding Decisions with Diverse Bidding Models

In this section, bidding decisions from some retail companies within 1000 learning rounds are compared. The bidding prices in different learning rounds are presented in Figure 7. When focusing on each subfigure in Figure 7, it can be seen that the selected probability of each bidding decision continuously changes. Taking S21 as an example, the bidding points are dispersed in the vertical direction in Figure 7d, indicating that the selected probabilities of each strategy in the early-stage bidding rounds are relatively close to each other. After a few rounds of learning, the bidding prices gradually converge at 0.416 RMB/kWh, indicating that the probability of this bidding price is comparatively high. However, the bidding decisions with lower probabilities also have opportunities to be selected because events with low probabilities are still possible events, which makes the simulation market uncertain and dynamic.

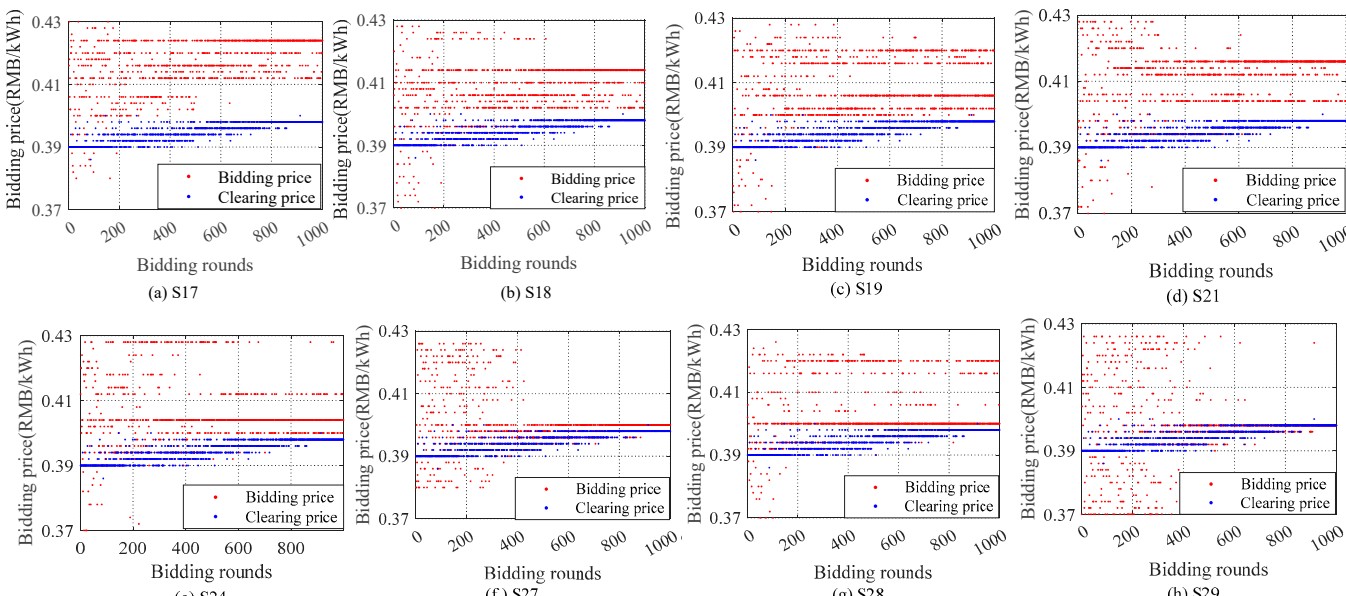

**Figure 7.** Bidding prices of power retail companies.

Four pairs of retail companies were selected to show the results, as shown in Table 2. Considering that the studied MRL system is complicated with many different agents and influencing factors, some retail companies will eventually converge on more than one price. In this paper, we present these prices in the order of their probabilities, from high to low.

**Table 2.** Overview of the comparison-in-pair of diverse bidding decision-makings.

| Example | Differences | Results | |
|---|---|---|---|
| | | **Bidding Results** | **Convergence Price (RMB/kWh)** |
| 1 | Different operation objectives | S29: max profit → Figure 7h | S29: 0.398 |
| | | S24: maximize market share in kWh → Figure 7e | S24: 0.404 |
| 2 | Different operation objectives | S17: defeat opponent S18 → Figure 7a | S17: 0.398 |
| | | S18: max profit → Figure 7b | S18: 0.424 |
| 3 | Different profit objectives | S27: fixed-delta-price: goal price + 0.02 RMB/kWh → Figure 7f | S27: 0.414 |
| | | S28: fixed-delta-price: goal price − 0.02 RMB/kWh → Figure 7g | S28: 0.400/0.42 |
| 4 | Different profit-sharing mode | S21: guaranteed price with a fixed-percentage sharing and interval fee-charging → Figure 7d | S21: 0.416 |
| | | S19: guaranteed price a with fixed-percentage sharing mode → Figure 7c | S19: 0.406/0.42/0.416 |
| | | S29: fixed-percentage sharing → Figure 7h | S29: 0.398 |

- S29 and S24: The bidding results of S29 and S24 are presented in Figure 7e,h, respectively. The results show that S24 finally converges at 0.404 RMB/kWh and S29 finally converges at 0.398 RMB/kWh. S29′s objective is to maximize profit, and thus, it bids a comparatively lower price to reduce the cost. At the same time, the prices bid by S29 in most rounds are almost equal to the clearing prices because the agent tends to guarantee it can be cleared. Different from S29, S24's bidding eventually converges to 0.404 RMB/kWh, which is consistently higher than the market clearing price, which allows the electricity to be successfully cleared with a high probability.

- When observing the bids in different rounds, both S24 and S29 converge within 300 rounds of learning, showing fast convergencies compared to the other retail companies. The prices bid by S29 are highly different in the early rounds, i.e., some are very high and some are very low. After MRL training, S29 finally converges its bids at 0.398 RMB/kWh. The reason accounting for this result is that the expected profit of S29 is 2 million RMB, and the final bidding price around the marginal price can lead to reaching such an objective. As for S24, it does not bid low prices to avoid bidding failure.

- S17 and S18: The bidding results of S17 and S18 are presented in Figure 7a,b, respectively. Since the main objective of S17 is to defeat S18, the bidding prices of S17 are always a little higher than the bidding prices of S18. As for the bidding process, S17 explores less space because it does not need to try the low price.

- S27 and S28: The bidding results of S27 and S28 are presented in Figure 7f,g, respectively. The results show that S27 converges rapidly to 0.400 RMB/kWh, while S28 converges to two prices, i.e., 0.400 RMB/kWh and 0.42 RMB/kWh, while the former is the same as S27 and has a larger probability. Both S27 and S28 aim to expand the market in kWh, and their profit-sharing modes are the same as the fixed-delta-price mode. The only difference between S27 and S28 in the case settings is that they have different profit expectations. Specifically, S27 has a minimum profit expectation of 2 million RMB, while S28 would rather expand on the market by sacrificing a maximum loss of 2 million RMB. As a result, the bidding decisions of S28 converge to a higher price with a small probability.

- S19, S21, and S29: The bidding results of S19, S21, and S29 are presented in Figure 7c,d,h, respectively. The results show that S19 converges rapidly to three prices, 0.406,

0.42, and 0.416 RMB/kWh; S21 converges to 0.416 RMB/kWh; and S29 converges to 0.398 RMB/kWh. The unique difference in the case settings of the three retail companies is the profit-sharing mode. Their objectives are to maximize the profit, and the different profit-sharing modes lead to differences in the bidding decisions. S29 converges to 0.398 RMB/kWh around the market clearing price. It does not need to guarantee the minimum profit for the end users, and the total profit will be fully shared between the retail companies and the end users. In another sentence, the greater the total profit, the greater the profit of the power retail companies. As a result, it prefers to bid a lower price for higher profit. Different from S29, S19, and S21 guarantee the end users a trading price, and their profits mainly depend on the profit, excluding the guaranteed part and service fee. Therefore, they have little motivation to venture to bid a low price, which may lead to bidding failures, and tend to take a stable bidding strategy at a high bidding price. As a result, their bidding prices converge to approximately 0.406 RMB/kWh.

### 5.3. Market Clearing Results

The market clearing curves of the electricity market "before MRL" and "after MRL" are shown in Figure 8. Note that only the third segment with MRL training parts is presented in the figures. Generational bidding prices are the same because they are not trained with MRL, and retail companies' bidding prices around the market clearing prices change significantly after MRL. In this case, most retail companies increase the bidding prices after MRL. To use S11 and S12, which are not trained with MRL, as a benchmark, six and twelve retail companies bid cheaper than S11 and S12 before MRL, respectively, but only one and three bid cheaper than S11 and S12 after MRL. As a result, the market clearing price after MRL increased from 0.39 RMB/kWh to 0.398 RMB/kWh, and the clearing quantities increased from 4.9 billion kWh to 5 billion kWh. The increase in market clearing prices indicates that the internal competition among retail companies is more intense when applying reinforcement learning. However, it should be noted that such a conclusion is highly related to market situations. In this case, the demand quantity from the retail companies is less than the offer quantity from the generation companies, and thus, the internal competition among the retail companies is theoretically not highly intensive. Nevertheless, reinforcement learning still increases the intensity between the retail companies.

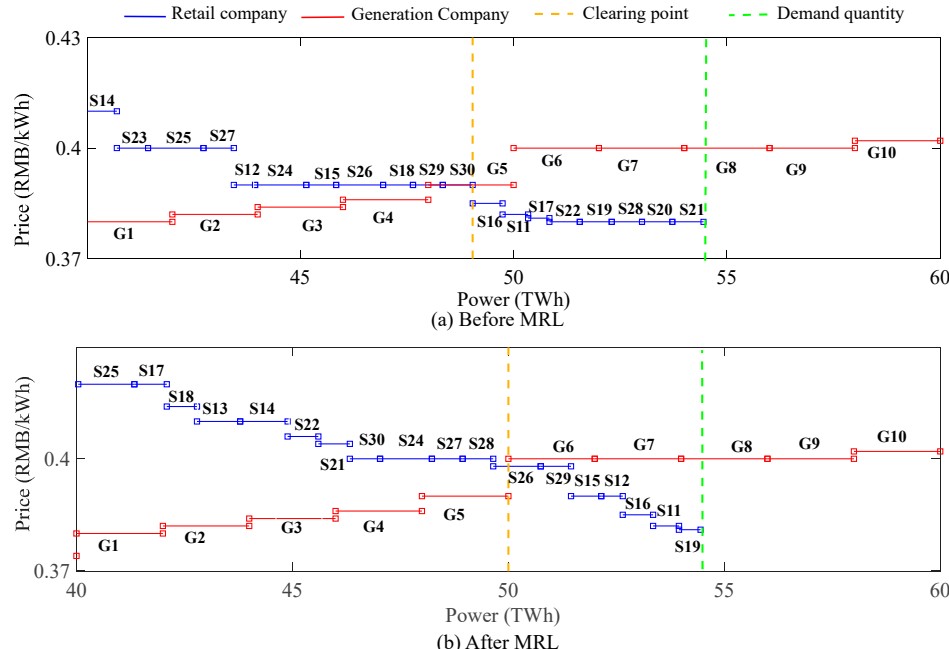

**Figure 8.** Market clearing results (1000 MRL rounds).

### 6. Conclusions and Discussions

In the field of agent-based electricity market modeling, most of the existing research focuses on the algorithm's development or assumes that power retail companies bid rationally to maximize profits, which is inconsistent with our observations in the real market. To fill the gap, this paper first establishes a model of diverse bidding decision-making among the power retail companies. The main conclusions include:

1. Three types of operation objectives and five types of profit-sharing modes are mathematically formulated. Three types of operation objectives include maximizing profit, market expansion in kWh or RMB, and defeating the opponent. Five types of profit-sharing modes include fixed-delta-price, fixed-percentage sharing, guaranteed price with a fixed percentage, service fee, and guaranteed price with fixed-percentage sharing and interval fee-charging modes.

2. The proposed models can effectively reflect the differences in the bidding decisions regarding their objectives and profit-sharing modes, which can finally lead to different bidding prices. In detail, retail companies with diverse bidding decision-making will have differences in the action space and converged pricing. Retail companies with the objective of maximizing profit will tend to bid lower prices, while those with the objective of maximizing market share will tend to bid higher prices. Comparing the results of different profit-sharing modes, the ones with guaranteed prices or service-fee modes tend to bid higher prices, while the ones with fixed-percentage modes are inclined to bid lower prices. In addition, the ones with higher profit expectations will bid higher prices. Moreover, due to the complicated system with multiple agents and diverse bidding models, the equilibrium conditions become complex. Some retailers tend to converge on several different bidding decisions, especially when these decisions will not largely influence their values of the objective functions.

3. A complete MRL-based electricity market simulation model is established, and simulation research is carried out in this paper. The case is well-designed to compare the impacts of one-by-one parameters of the retail companies while keeping the market environment exactly the same. Simulation results show that the internal competition among retail companies after MRL is more intense and will lead to an increase in market clearing prices. When the demand quantity from the retail companies is less than the offer quantity from the generation companies, the market clearing quantity tends to increase after MRL, showing positive effects for the agents.

Future work aims to enhance the proposed MRL-based model in three directions. The first one lies in improving the convergence speed and reducing the convergence difficulties of the proposed method. In fact, the MRL-based method is faced with the convergence problem when increasing the number of participants and enriching the bidding models of the participants. Although it is not mandatory or necessary to reach convergence when studying the process, and not every market will ideally reach equilibrium, convergence is an important aspect of such a study. Therefore, the simulation efficiency of the MRL-based method needs to be further enhanced. The second one lies in considering the interactions between the generation companies and the retail companies instead of assuming the generation companies use static bidding strategies. The third one lies in introducing new algorithms, e.g., deep RL, into the problem of exploring new findings.

**Author Contributions:** Conceptualization, Y.W., Methodology, Y.W. and C.L. Software, C.L. Validation, Y.W. Writing, Y.W., W.Y. and C.L. Funding acquisition, Y.W. and L.L. All authors have read and agreed to the published version of the manuscript.

**Funding:** This research was funded by State Grid Corporation of China with the project "Research on Multi-level Outage Planning Optimization and its Coordination Technology with Long-term Dispatching Decision" (No. 5108-202155042A-0-0-00).

**Institutional Review Board Statement:** Not applicable.

**Informed Consent Statement:** Not applicable.

**Data Availability Statement:** The simulation data is given in the Appendix A.

**Conflicts of Interest:** The authors declare no conflict of interest.

**Appendix A**

**Table A1.** End-user regulated prices [54].

| Type | | | Per kWh Price (Yuan/kWh) | | | | |
|---|---|---|---|---|---|---|---|
| | | | <1 kV | 1–10 kV | 35 kV | 110 kV | 220 kV |
| Residential end users | | Normal time | 0.5953 | – | – | – | – |
| | | Valley time | 0.3153 | – | – | – | – |
| Industrial or commercial end users | (plan A: pay as you go) | Peak time — July, August, September | 1.1739 | 1.1494 | 1.1250 | – | – |
| | | Peak time — Other | 1.1053 | 1.0823 | 1.0594 | – | – |
| | | Normal time | 0.7416 | 0.7266 | 0.7116 | – | – |
| | | Valley time | 0.4603 | 0.4514 | 0.4426 | – | – |
| | (plan B: package) | Peak time — July, August, September | – | 1.0181 | 0.9936 | 0.9692 | 0.9529 |

**Table A2.** Parameter settings for the simulation.

| NO. | Self Study | Power Needm (GWh) | Operation Objective | Profit-Sharing Mode | Sharing Percentage | Profit(+)/ Loss(-) Goal (Million RMB) | Fixed-Delta-Price (RMB/kWh) | Guaranteed Price (RMB/kWh) | Recency Factor (R) | Empirical Factor (E) | Service Fee (RMB/kWh) |
|---|---|---|---|---|---|---|---|---|---|---|---|
| S1–S10 | No | 20 | - | - | - | - | - | - | - | - | - |
| S11 | No | 60 | Max profit | Fixed percentage | 0.1 | - | - | - | - | - | - |
| S12 | No | 50 | Max profit | Fixed percentage | 0.1 | - | - | - | - | - | - |
| S13 | No | 100 | Market expansion in kWh | Fixed-delta-price | - | - | - | - | - | - | - |
| S14 | No | 110 | Market expansion in kWh | Fixed percentage | 0.5 | - | - | - | - | - | - |
| S15 | No | 70 | Market expansion in RMB | Fixed percentage | 0.5 | - | - | - | - | - | - |
| S16 | No | 70 | Market expansion in RMB | Fixed-delta-price | - | - | - | - | - | - | - |
| S17 | No | 50 | Defeat opponent (S18) | Fixed percentage | 0.5 | - | - | - | - | - | - |
| S18 | Yes | 70 | Max profit | Fixed percentage | 0.5 | +5 | - | - | 0.02 | 0.02 | - |
| S19 | Yes | 75 | Max profit | Hybrid: fixed-delta-price and fixed percentage | 0.1 | - | - | 0.39 | 0.02 | 0.02 | - |
| S20 | Yes | 71 | Max profit | Hybrid: fixed percentage and service fee | 0.5 | - | - | 0.39 | 0.02 | 0.02 | 0.12 |
| S21 | Yes | 73 | Max profit | Guaranteed price with a fixed percentage sharing and interval fee-charging | 0.1 | - | - | 0.4 | 0.02 | 0.02 | 0.12 |

**Table A2.** *Cont.*

| NO. | Self Study | Power Needm (GWh) | Operation Objective | Profit-Sharing Mode | Sharing Percentage | Profit(+)/ Loss(-) Goal (Million RMB) | Fixed-Delta-Price (RMB/kWh) | Guaranteed Price (RMB/kWh) | Recency Factor (R) | Empirical Factor (E) | Service Fee (RMB/kWh) |
|---|---|---|---|---|---|---|---|---|---|---|---|
| S22 | Yes | 71 | Max profit | Guaranteed price with a fixed percentage | 0.1 | - | - | 0.45 | 0.02 | 0.02 | - |
| S23 | Yes | 73 | Max profit | Fixed percentage | 0.5 | +2 | - | - | 0.1 | 0.02 | - |
| S24 | Yes | 120 | Market expansion in kWh | Fixed-delta-price | - | - | 0.02 | - | 0.02 | 0.02 | - |
| S25 | Yes | 130 | Market expansion in kWh | Fixed-delta-price | - | -54.9 | 0.02 | - | 0.02 | 0.02 | - |
| S26 | Yes | 110 | Market expansion in kWh | Fixed-delta-price | - | -44 | 0.02 | - | 0.02 | 0.02 | - |
| S27 | Yes | 71 | Market expansion in RMB | Fixed-delta-price | - | +2 | 0.02 | - | 0.02 | 0.02 | - |
| S28 | Yes | 71 | Market expansion in RMB | Fixed-delta-price | - | -2 | 0.07 | - | 0.02 | 0.02 | - |
| S29 | Yes | 70 | Max profit in RMB | Fixed percentage | 0.1 | +2 | - | - | 0.02 | 0.02 | - |
| S30 | Yes | 130 | Max profit in RMB | Fixed percentage | 0.5 | +2 | - | - | 0.02 | 0.02 | - |

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
