# Peer review of "MRL-Based Model for Diverse Bidding Decision-Makings of Power Retail Company in the Wholesale Electricity Market of China"

_axioms, doi:10.3390/axioms12020142_

Round 1

Reviewer 1 Report

The authors develop a 11 multiagent reinforcement learning (MRL)-based model to simulate the diverse bidding decision-makings of power retail company in the wholesale electricity market and also tried it with 30 retail companies.

 The authors should give references for the following expression: “In fact, power retail companies are mostly smaller in capacity, higher in flexibility, and more dispersed located compared to generation companies. Their bidding decisions are diverse and affected by many factors, which is not completely rational.”

 The regulations for power retail companies and the markets which power retail companies operate in are not the same in each country in the world. Therefore, the authors make some general assumptions for their models.

 There are referencing problems in the study. For example, the authors wrote the following expressions without any references: “This paper adopts the PAC pricing scheme, which is also the most common clearing mechanism.” I think that the paper needs a serious review for referencing.

 The authors should emphasize the advantages and disadvantages of MRL-based model from other alternatives and clearly write their motivation about use of MRL.

 The authors should clearly write where 30 retail companies are located.

 The authors should reconsider about the title of the paper. This is a general model or designed for China.

 The implications of the case study should be improved and future research directions should be given for the researchers.

Author Response

We highly appreciate the review feedback and we have carefully revised the paper addressing the concerns raised in the reviews. Please see the attachment for the response.

Reviewer 2 Report

The paper is interesting and important. I suggest more elaboration on the power industry structure and what makes this research unique, several articles have dealt with the same issues in China. moreover, the paper should address the bidding process from a known Game Theory approach: see for example Dong-Joo Kang, Balho H. Kim, Don Hur, Supplier bidding strategy based on non-cooperative game theory concepts in single auction power pools, Electric Power Systems Research, Volume 77, Issues 5–6,2007,

Author Response

Thank you for your positive feedback and helpful comments! We have carefully revised the paper. Please see the attachment for the response.

Round 2

Reviewer 1 Report

The authors have sufficiently addressed my critics. I suggest "accept in present form".

Congratulations 

Author Response

Thank you a lot for your agreement!